# Characterization of Counter-Surface Substrates for a Laboratory Abrasion Tester (LAT100) Compared with Asphalt and Concrete to Predict Car Tire Performance

Marzieh Salehi [1,*], Jacques W. M. Noordermeer [1,*], Louis A. E. M. Reuvekamp [2] and Anke Blume [1]

1   Department of Elastomer Technology and Engineering, University of Twente, P.O. Box 217, 7500 AE Enschede, The Netherlands; a.blume@utwente.nl
2   Apollo Tyres Global R&D B.V., Colosseum 2, 7521 PT Enschede, The Netherlands; louis.reuvekamp@apollotyres.com
*   Correspondence: m.salehi@utwente.nl (M.S.); j.w.m.noordermeer@utwente.nl (J.W.M.N.)

**Abstract:** Tire performance is determined based on the interaction between the tire and the road as a counter-surface, and is of the utmost importance for driving safety. When studying tire friction and abrasion, the characteristics of the roads/counter-surfaces are crucial. The excitations on the tire come from the road asperities. A proper characterization of the counter-surface texture is, therefore, an absolute necessity in order to optimize tire performance. The present study provides the required knowledge over the counter-surfaces employed as common substrates in a Laboratory Abrasion Tester (LAT100), which are typically based on embedded corundum particles for dry/wet friction and abrasion experiments. All surfaces are scanned and characterized by laser microscopy. The surface micro and macro roughness/textures are evaluated and compared with asphalt and concrete as the real roads by power spectral densities (PSD). The reliability of the high-frequency data based on the device type should be considered carefully. The reliable cut-off wavenumber of the PSDs is investigated based on image analyses on the range of tested frequency for micro and macro textures obtained by optical scanning devices. The influence of the texture wavelength range on the rubber−surface interaction is studied on a laboratory scale.

**Keywords:** tire friction; abrasion; surface characterization; power spectral density (PSD)

## 1. Introduction

Tire grip or traction, as a concept that describes the grasp and interaction between the tire and the road, is crucial for safety. Proper tire grip provides a good level of handling, which is a prerequisite for vehicle steering in various driving states, such as cornering, braking, and accelerating. The tire grip is the result of the generated frictional forces in the aforementioned driving states, which are created by tire slippage in the contact patch. The eventual contact of a full tire with the ground is the rubber tread compound. In order to optimize tire grip/traction, abrasion, and slip properties by improving the properties of these rubber materials, there is a continuous drive to find methods that allow for better simulation of the tribological characteristics of the tread compounds in early development stages in a laboratory environment. Full-scale tire tests are enormously sophisticated, time-consuming, and costly. It is, therefore, more sustainable and highly desirable to predict tire performance in a laboratory environment before having to manufacture a full tire. When studying tire performance, the characteristics of the roads/counter-surfaces are of crucial; the excitations on the tire come from the road asperities. It has always been a challenge to employ a realistic surface in a laboratory environment to evaluate the rubber tribological properties [1–9]. A proper characterization of the counter-surface texture is, therefore, an absolute necessity in order to obtain insight into tire performance in comparison with the real road surface. Sandpaper P120 is one of the common counter-surfaces used for the evaluation of full-scale tire properties in indoor testing, like so-called

Flat track measurements, or for the determination of rubber tribological properties with apparatuses, such as the Laboratory Abrasion Tester (LAT100) and Lambourn Abrasion Tester. The commonly used LAT100 counter-surface discs are composed of corundum particles employed for the evaluation and prediction of the tire performance [10–15]. In our previous studies [2,12,13], LAT100 was employed as a laboratory tribometer, where strong correlations were obtained between the laboratory results and the tire data only on a specific counter-surface substrate. The correlations were acquired experimentally and were verified and supported with statistical analyses [12,13] and modeling [16]. The counter-surface exploited for laboratory measurements was LAT100 corundum disc and was seemingly different from the common tire−road surfaces. The present study supports the previous work by obtaining insight into the employed substrates for laboratory and road testing; the LAT100 counter-surfaces were analyzed in comparison with typical asphalt and concrete using power spectral density (PSD) to enhance the explanations for the acquired correlations. This provides a better picture of the acquired correlation and highlights the potential characteristics of the laboratory counter-surface substrate to measure and evaluate tire performance. It can create a new cost- and time-effective perspective for tire material development in a laboratory environment, rather than having tested tread compounds by building full-scale tires.

### 1.1. Classification of Road Surface Texture

The texture or roughness of the road counter-surface plays a dominant role in tire performance such as friction, wear, noise, rolling resistance, and splash/spray on wet surfaces. Various textures contribute differently to friction components and involve different mechanisms for abrasion. The Permanent International Association of Road Congress (PIARC) has defined a scale based on the texture wavelength of the contact surface to classify the characteristics and the impact of roughness on tire performance. Figure 1 shows the influence of the texture wavelength on the tire track interaction/performance. The micro-texture (below about 0.5 mm) is of fundamental importance on dry roads, and interacts directly with the tire friction on a molecular scale. The other tire performance characteristics are illustrated in Figure 1 [17].

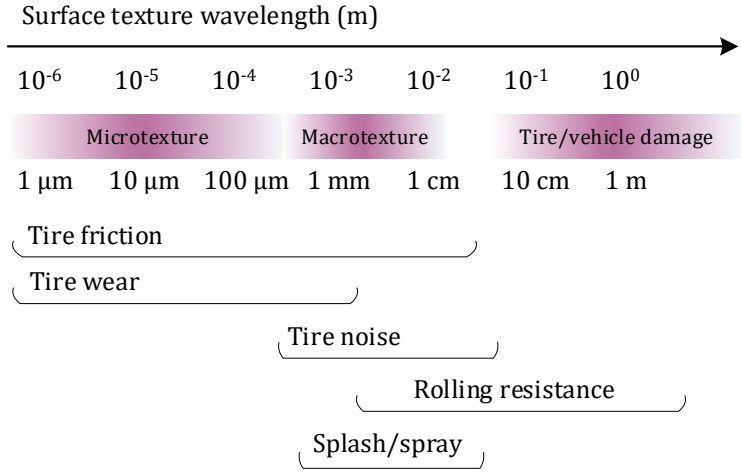

**Figure 1.** Influence of the texture wavelength on the tire–surface interaction, redrawn based on [17].

A locally specified anti-skid top layer is usually applied on asphalt roads with a surface is gritty and resistant to abrasion. The road surface is commonly engineered in such a way that by adjusting the micro and macro texture, the tire performance is balanced for a wide range of tires (see Figure 1). Macro texture is typically considered on a scale of 0.5–20 mm and micro texture refers to asperities less than 0.5 mm across the surface. The irregularities of the road surface that are lower than 10 mm (100 cycle/m wavenumbers), and their impacts are largely absorbed by the tire before they reach the springs of a car

suspension system (see Figure 2). The surface profile can be characterized in positive and negative structures [18] (as depicted in Figure 3).

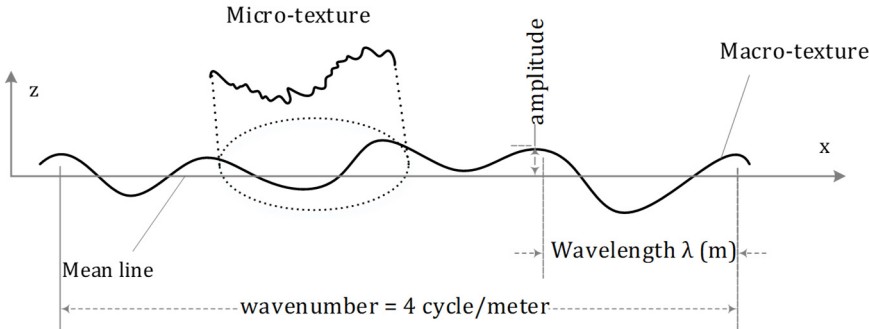

**Figure 2.** Macro- and micro-texture representation and wave terminology.

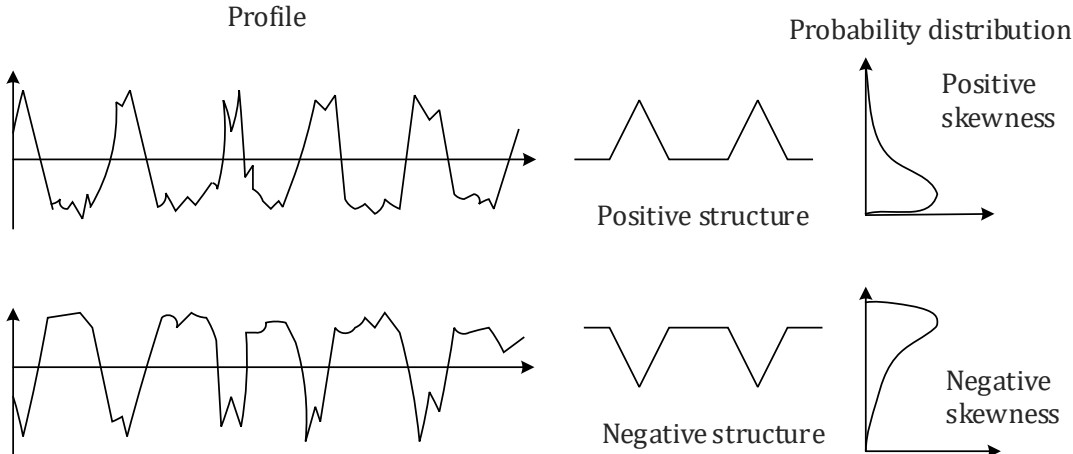

**Figure 3.** Positive and negative structures and distributions resulting in positive or negative skewness of the counter-surface, redrawn based on [18,19].

*1.2. Characterization of Texture*

Numerous roughness parameters are available that can be extracted from texture measurement methods, which can be carried out in the lab or outside, such as contactless optical microscopic methods and contact-type profile measuring instruments. The most basic and common roughness parameters are $R_a$, $R_q$ which are calculated from the amplitudes of the asperities. The amplitude parameters characterize the surface based on the height deviations (in $z$ direction in Figure 2) of the roughness profile from the mean line; the mathematical definitions are as follows:

$$R_a = \frac{1}{L} \int_0^L |z(x)| dx \tag{1}$$

$$R_q = rms = \left( \frac{1}{L} \int_0^L z(x)^2 dx \right)^{1/2} \tag{2}$$

where $R_a$ $(m)$ is an arithmetic average of the absolute values of the height $z$ $(m)$ of the asperities relative to the mean line over a distance $x$ $(m)$ on the length of measurement $L$, and $R_q(m)$ is the root mean square ($rms$) of the surface heights. $R_{sq}$ is another useful parameter for the shape of the asperities: the skewness can indicate a positive or negative

structure of the counter-surface (Figure 3) [20]. Two surfaces with the same $R_q$ can have different shapes [19].

$$R_{sq} = \frac{1}{LR_q^3} \int_0^L z(x)^3 dx \qquad (3)$$

A surface with a large $R_a$ value or a positive $R_{sq}$ usually produces high friction and wears quickly. More parameters should be considered, namely the form and waviness of both amplitudes and the frequency of the asperities, as well as their slopes and spacing's. A finite number of surface parameters to describe the real surface geometry is mainly scale-dependent and provides only limited information [19].

In 1971, Nayak [21] modeled rough surfaces as two-dimensional isotropic Gaussian random processes, and analyzed them with the aid of random process theory. The surface statistics such as the distribution and density of heights, and the RMS height and RMS slope are related to the power spectral density (PSD) of the surface profile; it is a combination of several important surface parameters in one tool. The use of random process analysis has been reported as a breakthrough in recent years, based on the idea of the autocorrelation function first used in 1946 [22]. The autocorrelation function describes the height similarity of two points on the surface at a distance $\tau$.

$$A(\tau) = \frac{1}{L} \int_0^L z(x)z(x+\tau)dx \qquad (4)$$

which At $\tau = 0$, $A(\tau) = R_q{}^2$. PSD contains statistical characteristics of the surface topography, regardless of the choice of the particular scan size. It decomposes the surface into a superposition of numerous sine and cosine signals of different amplitudes and wavelengths. Using the Fourier transform technique, a wavelength with a specific amplitude can be converted to a wavenumber with a specific power. The wavenumber is the number of cycles per meter representing the frequency of the asperities. The term "power" originated from electronic engineering and is a common practice to measure the power content associated with different wavelengths rather than just the amplitudes [18]. Mathematically, PSD is the Fourier transform of the autocorrelation function of the signals with different spatial frequencies (wavevectors) [22]. There are variations in the mathematical definitions and units of PSD [7]. The 2D power spectrum $C_{2D}(\boldsymbol{q})$ in the unit of $m^4$ holds as follows [23]:

$$C_{2D}(\boldsymbol{q}) = \frac{1}{(2\pi)^2} \int h(\boldsymbol{x})h(\boldsymbol{0})e^{-i\boldsymbol{q}\boldsymbol{x}}d^2x \qquad (5)$$

where the wavenumber $q = 2\pi/\lambda$ is the magnitude of the wavevector $\boldsymbol{q}$ (m$^{-1}$) with the wavelength of $\lambda$ (m), and $h(\boldsymbol{x})$ is the height coordinate at the point $\boldsymbol{x} = (x, y)$. The bold letter is considered as a vector. The roughness amplitude RMS can be written as follows [8]:

$$R_q^2 = 2\pi \int_{q_0}^{q_1} q\, C_{2D}(q)dq \qquad (6)$$

where $q_0$ is the small cut-off wavevector (long roll-off wavelength) and $q_1$ is the large cut-off wavevector (short wavelength). The slope and curvature RMS will be the first and second derivatives of Equation (6).

To characterize a fractal surface, an understanding of the concepts of self-similarity and self-affinity is required. A fractal is either self-similar or self-affine. The physical appearance of a fractal is the same, regardless of the scale at which it is observed. In mathematics, self-affinity is a feature of a fractal whose pieces are scaled by different amounts in the $x$- and $y$-directions; the $y$-direction must be rescaled by $a^H$ to preserve the $x$-direction defined by $a$, where $H$ is a value between 0 and 1 [24] (described later). In nature, a variety of surfaces are associated with self-affine fractal scaling, defined probably first by Mandelbrot in terms of fractional Brownian motion. Self-affine surfaces in three dimensions can be distinguished from self-similar ones by an asymmetry in the scaling behavior orthogonal

to the surface $xy$-plane, i.e., $z$-axis [25]. For a self-affine surface, the power spectrum has a power-law behavior [8]:

$$C_{2D}(q) \sim q^{-2(H+1)} \tag{7}$$

where the Hurst exponent $H$ is related to the fractal dimension by $D_f = 3 - H$, which is usually <2.3. A fractal dimension $D_f \approx 2.2$ ($H \approx 0.8$) is typical for asphalt and concrete roads [8]. Theoretically, it is possible to convert a 1D PSD obtained with a linear roughness measurement, such as a stylus to a 2D PSD. Figure 4 depicts typical 2D PSD and 1D PSD graphs that differ in the slope values. The relationship between a 1D and 2D PSD for an isotropic surface is as follows [7]:

$$C_{2D}(q) \approx \frac{\pi}{q} C_{1D}(q) \tag{8}$$

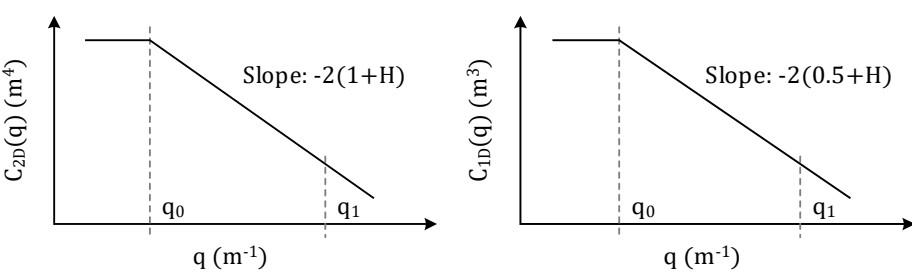

**Figure 4.** Theoretical log–log representations of 2D- and 1D-PSD for an isotropic self-affine fractal surface; with $q_0$ as the small cut-off wavevector (long roll-off wavelength) and $q_1$ as the large cut-off wavevector (short wavelength).

The cut-off wavevectors should be taken from reliable data, which depend on the accuracy of the measurement [7]. There are challenges for accurate determination of the 2D-PSD measured with optical microscopy due to various types of surface topography techniques and resolution limits. Deviation from a 1D stylus linear scanner and a 2D-optical scanner typically takes place in a higher frequency regime than $10^4$ m$^{-1}$ [26].

Engineering surfaces may also have anisotropic statistical properties, and therefore frictional properties depend on the direction of sliding. For surfaces polished or ground in one direction, contact mechanics depend on the radially averaged power spectrum [8,27].

## 2. Experimental

The test tracks typically employed for indoor and laboratory friction or abrasion experiments are corundum-based substrates. In the present study, the commonly used substrates for friction and abrasion measurements are studied. The detailed experiments of dry/wet friction tests are described in references [16,28]. The surfaces compared and investigated are as follows:

- Road samples: asphalt and concrete;
- Sandpaper grade P120;
- LAT100 counter-surface discs 60, 180, 320, and 180B;
- A comparably smooth surface, such as an A4 paper sheet.

The road sample provided by Apollo Tyres Global R&D was a typical dense asphalt prepared based on an the ABS braking test procedure according to ASTM F1649 for wet traction performance of passenger car tires. A 30 cm$^2$ tile of concrete was selected for the contact area measurements in the laboratory, which is comparable to the asphalt sample. The asphalt and concrete samples can be considered as typical test tracks of the existing ground in Europe used for tire testing [2]. Sandpaper P120 is a fine grade of silicon carbide with an average particle size of 125 µm, which is classified according to ISO 6344/FEPA Grit Designation. The employed LAT100 counter-surface discs were provided by VMI Holland B.V., Tire Industry Equipment. They are composed of electro-corundum

white $Al_2O_3$ powder bound in a ceramic binder with a weight ratio of 85/15%. They are classified according to ISO 525:2013(E), as given in Table 1, of which the size-fraction of the corundum-particles is the most important variable in the present context. Each disc code represents the disc grain size; a higher code number indicates a smaller corundum particle size. The corundum particles in the ceramic binder are pressed together and the disc surfaces are flattened by a grinding process using fine steel powder. The letter "B" for disc 180B indicates "blunt", which means that disc 180 was smoothened one step further in the finishing/flattening process using a grinding disc containing diamond powder. This smoothening process does not influence the grain size; however, it affects the surface structure and sharpness of the asperities, as shown in Figure 3. The corundum-based surfaces have been used in laboratory friction measurements in our previous studies to predict tire performance [12,13,29].

**Table 1.** LAT100 disc characterizations.

| Disc Designation | 60 | 180 | 320 | 180B |
|---|---|---|---|---|
| Disc code | EKW 60 SHARP | EKW 180 SHARP | EKW 320 | EKW 180 BLUNT |
| Average particle size (µm) | 225 | 60 | 32 | ↑ |
| Corundum sieve analysis (µm) | 0% > 425<br>Max. 30% > 300<br>Min. 40% > 250<br>Min. 65% > 212 | 0% > 125<br>Max. 15% > 90<br>Min. 40% > 63<br>Min. 65% > 53 | Max. of 52 *<br><br>Min. of 19 * | ⋮ Same as disc 180 ⋮<br><br>↓ |

\* Photosedimentometer analysis (µm).

Microscopic images of the experimental tracks were obtained with two different laser scanning microscopes at various resolutions in order to obtain the micro and macro textures of the counter-surfaces:

1.  A Keyence confocal laser scanning microscopy VK 970 for the indoor testing. A maximum scanning area of 2 cm$^2$ can be acquired by stitching and combining multiple measurements into one image. Three lens magnifications of 10×, 50×, and 100× at different resolutions were employed.
2.  A Stemmer imaging laser scanner with a Micro-Epsilon optoNCDT2300 triangulation laser sensor with a maximum height range of 20 mm. The XY-stage has a range of 100 mm × 100 mm and a maximum resolution of 10 µm/pixel. This device was utilized to assess the macro texture.

To compare both apparatus settings, the devices were evaluated at a specific surface. As the Keyence device has a certain limit regarding the sample dimension and weight, the Stemmer device was used as a non-destructive method for the counter-surface samples, which also enables scanning a larger area and providing information over the larger asperities. It should be noted that the measurements with a confocal scanning microscope cannot be compared accurately at different measurement conditions such as lens magnification, optical zoom magnification, measurement mode, scan settings, scan format (single, double), measurement pitch, XY calibration, and Z calibration.

## 3. Results and Discussion

Macro and micro textures can approximately be distinguished in a PSD graph, as depicted in Figure 5, for wavenumbers, which correspond to the defined wavelengths in Figure 1. The magnitude of $C(q)$ is associated with the power of the amplitudes of the asperities; the higher the $C(q)$, the larger the amplitude. Subsequently, the steeper the slope, the rougher the surface. In the current section, the macro and micro textures are compared utilizing the available laser scanners.

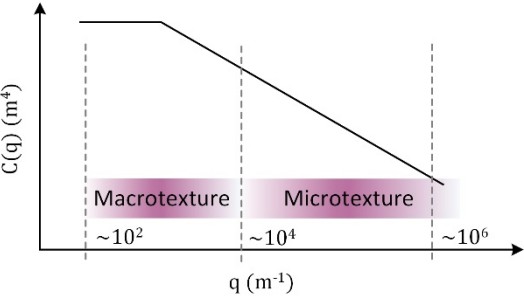

**Figure 5.** The distinction of macro and micro textures on the PSD graph.

### 3.1. Macro-Texture Characterization

In the first step, the Keyence and Stemmer scanners are compared by scanning the asphalt sample, which is known as a typical self-affine surface. The Keyence device with $10\times$ magnification provided an image by stitching $20 \times 20$ small images at 5524 nm/pixel to a final dimension of 24.9 mm $\times$ 17.8 mm, as shown in Figure 6. The Stemmer scanner provided a reciprocal linear scan of an area of 50 mm $\times$ 50 mm at the highest possible resolution of 10 µm/pixel. The 2D-PSD of both measurements are compared in Figure 7. PSD ends at the limits of the resolution of the scanners. The middle regions are in good agreement with the basic principles of a linear fit of the PSD; the region at high wavenumbers, which deviates from the line, is considered as an unreliable region, as specified in Figure 4, which is due to the outliers of the measurement (the spikes in Figure 6). Due to the larger area scan of 5 cm$^2$ by the Stemmer device, the PSD graph is extended to the lower wavenumbers. A Hurst exponent $H = 0.8$ for typical road asphalt is illustrated in Figure 7. Therefore, the Keyence and Stemmer scanners both give comparable results within the adjusted range of resolutions.

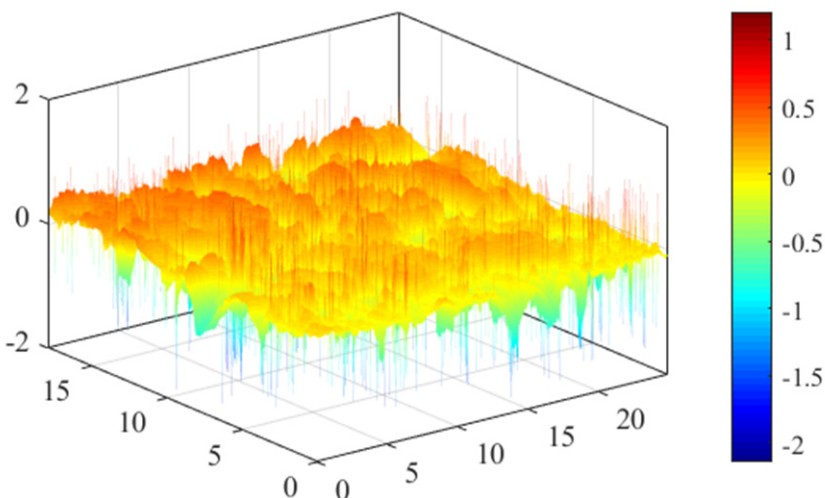

**Figure 6.** 3D view of the asphalt sample scanned by the Keyence device with a resolution of 5524 nm/pixel and the dimension of 24.9 mm $\times$ 17.8 mm; axes in mm.

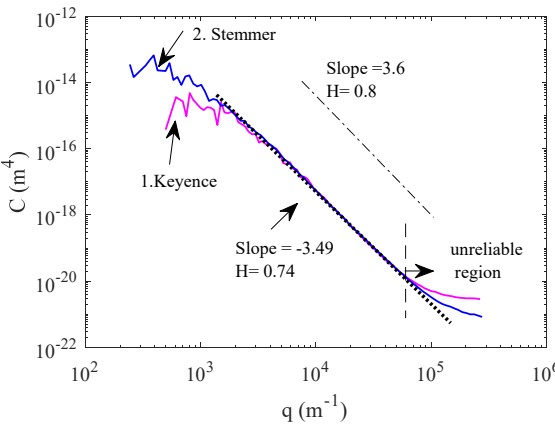

**Figure 7.** 2D-PSD graphs of the asphalt surface obtained with the Keyence and Stemmer laser scanners.

To evaluate the macro texture, all the counter-surfaces under study were characterized by the Stemmer scanner; the images were acquired at a resolution of 10 μm/pixel, the calculated 2D-PSDs are presented in Figure 8. Long-distance roll-off wavelengths were observed for LAT100 discs and sandpaper P120: at the beginning of PSD, $C(q)$ becomes invariant. Corundum particles produced by sintering and a sandpaper surface typically have this property of a long-distance roll-off wavelength [23]. The PSD graphs of the LAT100 discs are in accordance with the particle size distribution in Table 1; however, the nominally flat surfaces of LAT100 discs and sandpaper still have amplitudes shorter than the diameters of the particle sizes. By considering the reliable region for wavenumbers smaller than $6 \times 10^4$ m$^{-1}$ from Figure 7, two bumps can still be detected in the PSD graphs of the disc 180 and P120 surfaces. It is a point of discussion whether the data are trustworthy due to reaching the resolution limit. It should be noted that the nature of the corundum material compared to asphalt and concrete is shiny and more reflective, which could make the optical laser scanner techniques less effective for characterization. Lorenz and et al. [26] reported that the 2D PSD obtained by the contact-less optical method deviates at wavenumbers larger than $10^4$ m$^{-1}$ from 1D PSD assessed by a Stylus scanner. The reliability cut-off wavenumber for optical measurements is also elaborated in detail in Jacobs's work [7]. According to his explanation, even for concrete, the region between $10^4$ m$^{-1}$ and $10^5$ m$^{-1}$ is not reliable due to the small bend on the PSD line.

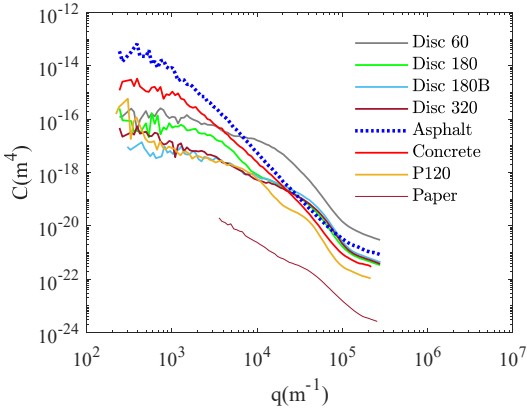

**Figure 8.** PSD graphs of all of the surfaces obtained by the Stemmer scanner at 10 μm/pixel resolution.

To clarify this, the image of LAT100 disc 180 was examined by pixel and image processing, as presented in Figure 9. Some pixels in a scanning line of order 10 μm shifted in the images during scanning measurements; this could occur due to environmental disturbances. The shifts were modified by removing the outliers and also with a Gaussian

filter with 1σ (standard deviation), which offered a cut-off frequency of $5 \times 10^4$ m$^{-1}$ (see Figure 9). The heart of the matter is that the region between $10^4$ m$^{-1}$ and $10^5$ m$^{-1}$ depends on the surface type and the quality of the image. However, disc 180 and P120 are surfaces with a particular particle size distribution and are polished/ground and tend to show asymmetric behavior. For instance, the average particle size 125 µm of P120 gives a $q = 2\pi/(2*125*10^{-6}) = 2.5 \times 10^4$ m$^{-1}$, where is the onset of the second bump, and it is followed by a slight depression. Overall, the counter-surface could be properly compared in the region lower than $q = 2.5 \times 10^4$ m$^{-1}$ in the macro texture, according to Figure 8.

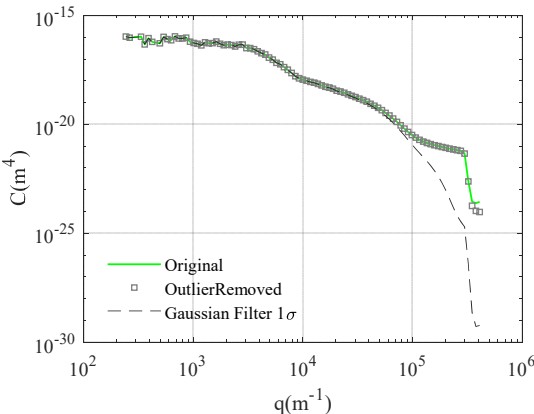

**Figure 9.** Image processing and filtering on the microscopic image of disc 180.

### 3.2. Micro-Texture Characterization

Second, microscopic images of all surfaces were scanned with the Keyence scanner. Figure 10 shows some examples, disc 180, sandpaper P120, and an A4 paper sheet and their 3D representations. Their 2D-PSD graphs are calculated and depicted in Figure 10. The resolution of the images is 276 nm/pixel, accordingly, the largest $q$ value on the plot should be of the order $10^7$ m$^{-1}$. However, again the reliability of the high-frequency data based on the device type should be taken with care. The PSD graphs of disc 180 and P120 surfaces start roughly from the same $C(q)$ value; as can be seen in Figure 11, the amplitude of the asperities is of the same order of magnitude (between 50 µm to 100 µm). The slope of the PSD graph for P120 is steeper than for disc 180 due to the size and distributions of the particles on the surface.

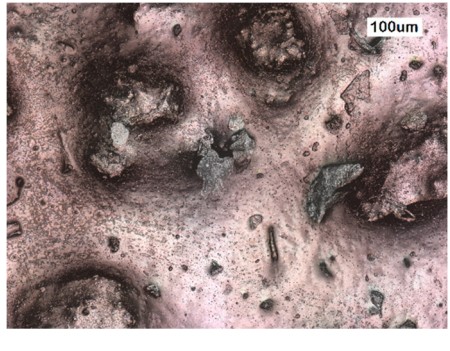 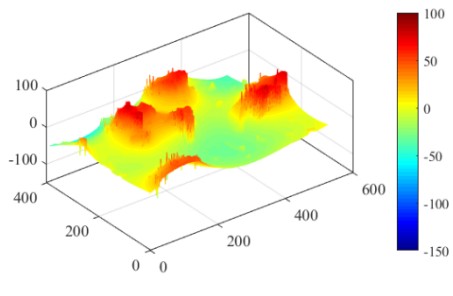

(**a**)

**Figure 10.** *Cont*.

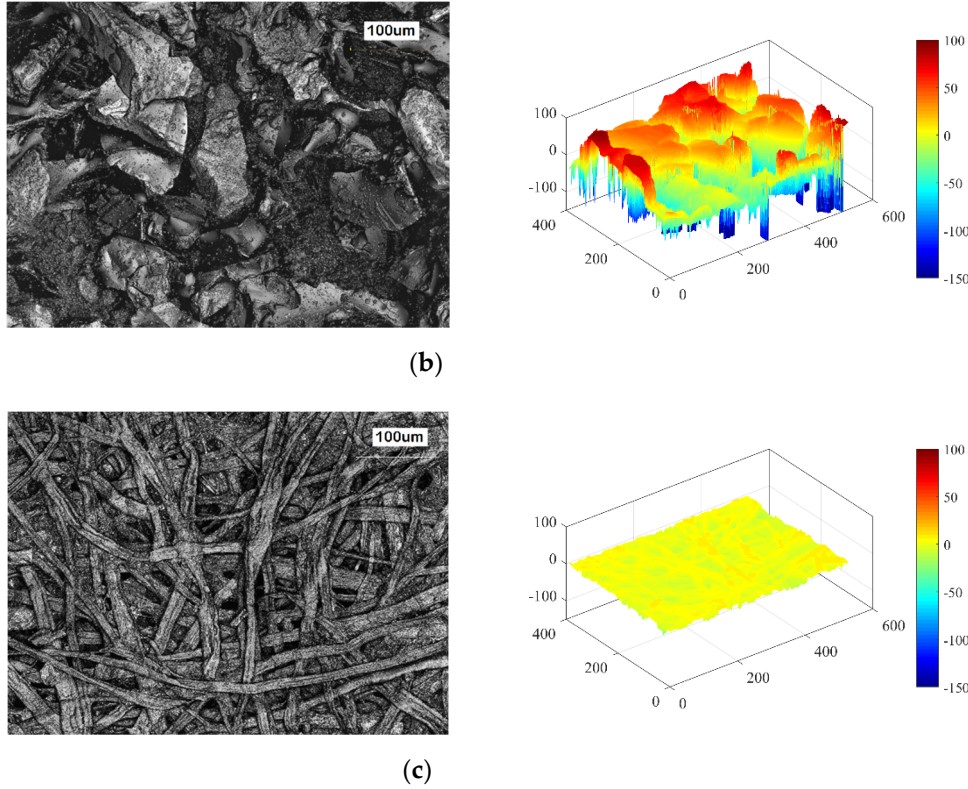

**(b)**

**(c)**

**Figure 10.** Microscopic images by the Keyence scanner at 50× magnification (**left**) and their 3D views (**right**) of (**a**) sandpaper P120, (**b**) disc 180, and (**c**) A4 paper, respectively; the scale of the images is 100 μm with the resolution of 276 nm/pixel; the axes scales are in μm.

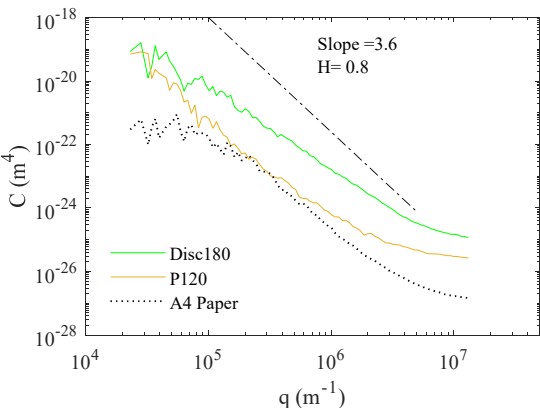

**Figure 11.** 2D-PSD of the microscopic images of Figure 10; sandpaper P120, disc 180 compared with an A4 paper sheet.

The microscopic images of the LAT100 discs obtained with the Keyence scanner at 10× magnification with 0.05 μm vertical pitch are illustrated in Figure 12; with an increasing disc code of 60–180–320, the grain asperities decrease in size. The blunting process of the disc 180B has resulted in a much smoother, flattened surface, in which the individual corundum particles can still be discerned to be of about the same size as in disc 180, but with the asperities reduced to about half the height and a lower RMS comparable to disc 320. The PSD graphs and roughness parameters are given in Figure 13 and Table 2, measured on an area of 1414 μm × 1080 μm.

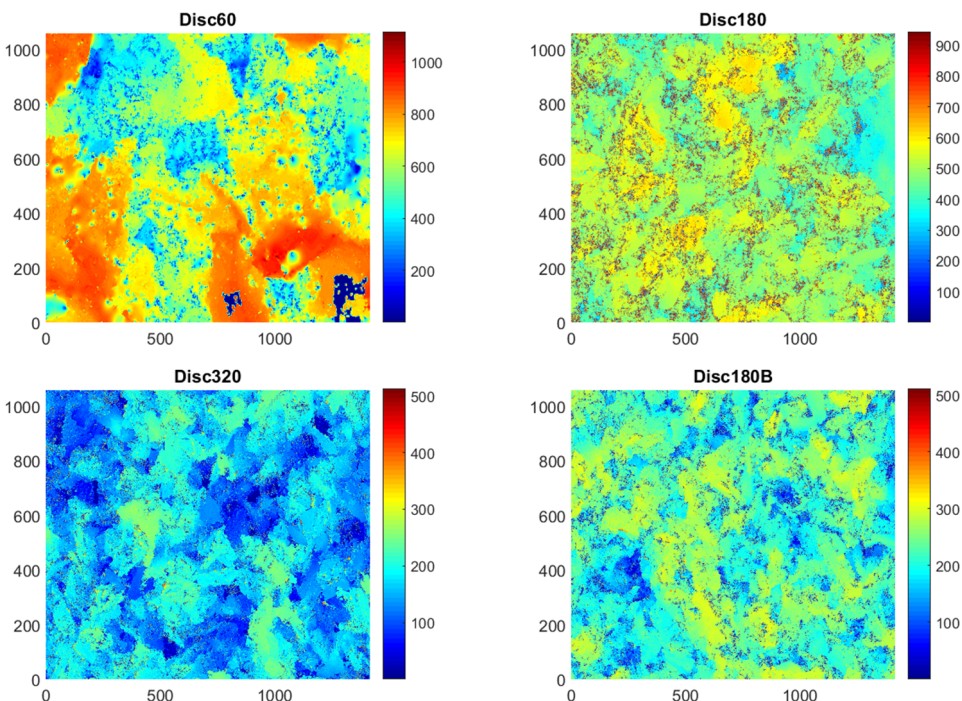

**Figure 12.** Roughness surface analyses with confocal laser scanning microscopy of the LAT100 discs employed; height numbers and scales in μm.

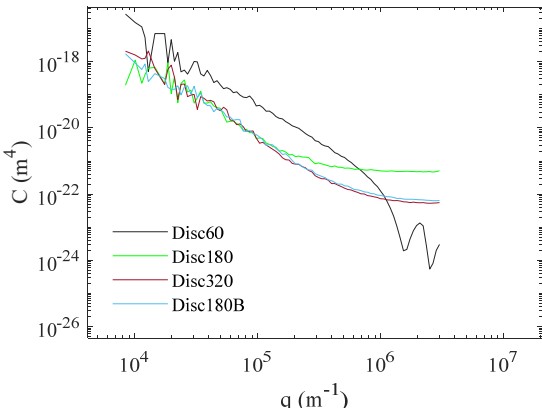

**Figure 13.** PSD graphs of LAT100 discs.

**Table 2.** Surface roughness parameters, measured with the Keyence scanner, $10\times$ magnification over an area of 1414 μm $\times$ 1080 μm. The notation S is being used rather than R because the measurements are done over a surface.

| Surface | Sa (μm) | Sq (μm) | Ssq (μm) |
|---|---|---|---|
| Disc 60 | 146 | 175 | −0.44 |
| Disc 180 | 47 | 65 | −1.37 |
| Disc 320 | 39 | 48 | −0.33 |
| Disc 180B | 35 | 44 | −0.56 |

### 3.3. Combined Macro and Micro Characterization: Overall Picture

The cut-off wavenumber $q_1$ commonly applied in calculations of the viscoelastic contribution of rubber friction on a road surface is around $10^6$ m$^{-1}$ [26,30]. This frequency typically marks the transition to the glassy behavior of rubber. To obtain the overall PSD, the measurement with the Stemmer scanner is combined with the Keyence. The concrete

surface is shown in Figure 14 and the 3D view in Figure 15; the combined PSD is represented in Figure 16. For $q$ above $10^6$ m$^{-1}$, deviation starts to occur from a straight line for the reason explained earlier.

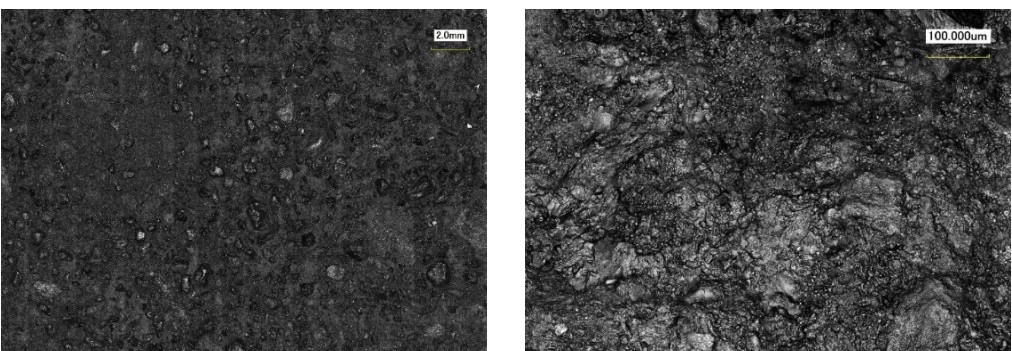

**Figure 14.** The concrete surface at two different magnifications of $10\times$ and $50\times$ by the Keyence device; scales are 2 mm and 100 µm.

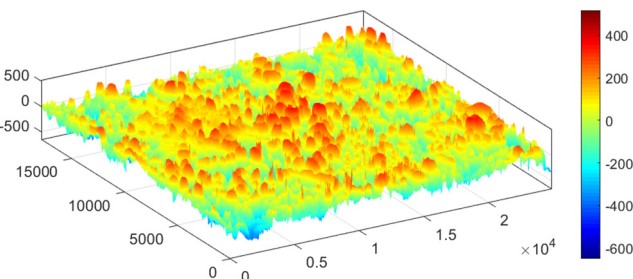

**Figure 15.** 3D view of the concrete surface, scale is µm.

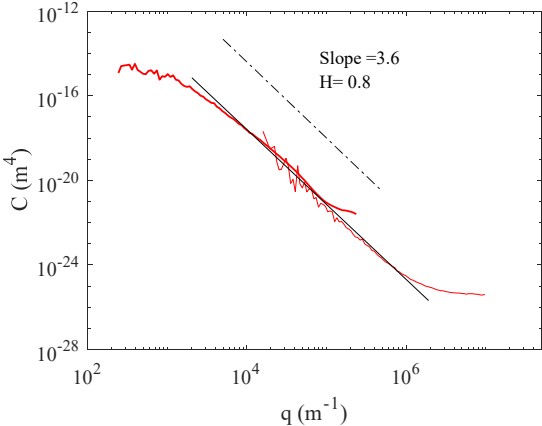

**Figure 16.** Combined PSD for the concrete surface.

Figure 17 depicts the overall PSD graphs for the LAT100 discs combined. The unreliable regions at higher frequencies are removed. The graphs could be extrapolated to a higher frequency, which is commonly done in a linear manner, based on other techniques like atomic force microscopy (AFM) [8,31]. However, above $10^6$ m$^{-1}$ is out of the scope of the current research.

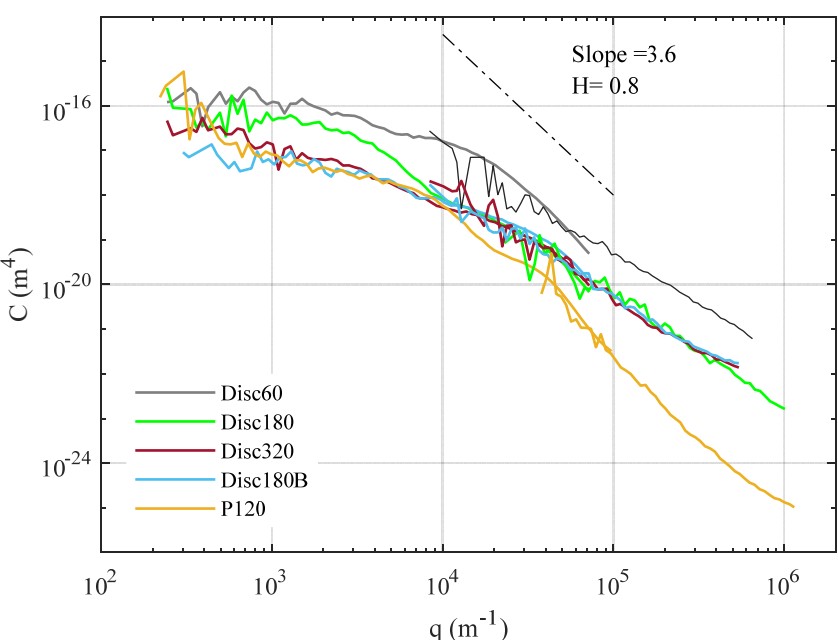

**Figure 17.** The combined PSD graphs of LAT100 discs.

## 4. Conclusions

Tire performance is the result of the interaction between the tire and the road as a counter-surface. In our previous study [2,12,13], LAT100 was employed as a laboratory tribometer, where strong correlations were obtained between the laboratory results and the tire data only on a specific counter-surface substrate. The correlations were acquired experimentally and were verified and supported with statistical analyses [12,13] and modeling [16]. The counter-surface exploited for laboratory measurements was LAT100 corundum disc 180 and seemingly different from common tire−road surfaces. The present study supports the previous work by obtaining insights into the employed substrates for laboratory and road testing. The counter-surfaces employed as common substrates in laboratory dry/wet friction and abrasion testers were characterized by optical methods utilizing laser scanners: Keyence and Stemmer. The surfaces' micro and macro roughness/structure were evaluated as an important element for the friction and abrasion. The power spectral densities (PSD) of the surfaces were analyzed and compared. The reliable cut-off wavenumbers of the PSDs were discussed. From the PSD graphs of the surfaces in comparison with asphalt, it can be deduced that:

- There is a shift upwards for LAT100 disc 60, which suggests the asperities with the same wavenumbers in comparison with other discs have larger amplitudes representing the micro texture of that surface.
- LAT100 disc 320 shows smaller amplitudes for macro texture compared to disc 180 and asphalt; however, the micro texture is in the same range as for discs 180 and 180B.
- The extra flattening for disc 180B affects only the macro texture and the structure or the shape of the asperities compared to disc 180. The micro texture is the same.
- Sandpaper P120 has a surface with a positive $R_{sq}$, while disc 180 has a negative one. It also has a higher $H$ exponent indicating a rougher surface than disc 180.

The in-depth study of the counter-surface substrates by PSD supplemented the explanations for the acquired correlations. Only disc 180 of LAT100 suggested a strong correlation for laboratory friction measurements with tire-road data in the previous studies [2,12,13]. The PSD graphs provided a better picture of the macro and micro texture of the counter-surface substrates for the acquired correlation, and highlighted the required characteristics of laboratory counter-surface substrates to evaluate tire performance. The influence of the texture wavelength range on the rubber−surface interaction could be

further studied on a laboratory scale. It opens a new cost- and time-effective perspective for tire material development in a laboratory environment, rather than having tested tread compounds by building full-scale tires.

**Author Contributions:** Conceptualization, M.S. and J.W.M.N.; methodology, M.S.; software, M.S.; validation, all; formal analysis, M.S.; investigation, M.S.; resources, A.B.; data curation, M.S.; writing—original draft preparation, M.S.; writing—review and editing, M.S. and J.W.M.N.; visualization, M.S.; supervision, J.W.M.N.; project administration, L.A.E.M.R.; funding acquisition, A.B. and L.A.E.M.R. All authors have read and agreed to the published version of the manuscript.

**Funding:** This research was funded by Apollo Tyres Global R&D B.V., Colosseum 2, 7521PT Enschede, The Netherlands: University of Twente project OFI no.: 20965331.

**Data Availability Statement:** The data presented in this study are available on request from the corresponding author.

**Acknowledgments:** The present research was financed by Apollo Tyres Global R&D B.V., Enschede, the Netherlands. The authors would like to express their gratitude to VMI group Holland B.V. for their support.

**Conflicts of Interest:** The authors declare no conflict of interest.

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
