# Peer review of "Characterization of Counter-Surface Substrates for a Laboratory Abrasion Tester (LAT100) Compared with Asphalt and Concrete to Predict Car Tire Performance"

_lubricants, doi:10.3390/lubricants10010008_

Round 1

Reviewer 1 Report

I appreciate the effort of this research, I mention that the laboratory work is interesting, but the interpretation of the data is a poor interpretation of the data.
On the other hand, there are things that are not accepted: the format must be respected, there are editing errors/errors and shortcomings.
I remember some of the typos, but they are many:
Aligned 54 Error! Reference
Aligned 58 Error! Reference source not found.
Aligned 64 Error! Reference source not found.
Aligned 68 Error! Reference source not found.
Aligned 70 Error! Reference source not found.
Paragraph 83 Error! Reference source not found.
Paragraph 89 Error! Reference source not found.
Paragraph 137 Error! Reference source not found.
 282 untitled figure
For example, references/comments on figures and tables in the article do not appear in the text.
Regarding the bibliography - there are very few new titles (from the last 5 years) !!
The conclusions are not very concise and do not highlight the strengths of the article.

Author Response

Reply to Reviewers ‘comments, Manuscript ID: lubricants-1503333

#Reply to review 1

I appreciate the effort of this research, I mention that the laboratory work is interesting, but the interpretation of the data is poor interpretation of the data. On the other hand, there are things that are not accepted: the format must be respected, there are editing errors/errors and shortcomings.
I remember some of the typos, but they are many:
Aligned 54 Error! Reference
Aligned 58 Error! Reference source not found.
Aligned 64 Error! Reference source not found.
Aligned 68 Error! Reference source not found.
Aligned 70 Error! Reference source not found.
Paragraph 83 Error! Reference source not found.
Paragraph 89 Error! Reference source not found.
Paragraph 137 Error! Reference source not found.
 282 untitled figure
For example, references/comments on figures and tables in the article do not appear in the text.

We thank the reviewer for the comments on the manuscript. We added some additional explanations (marked in red in the revised manuscript) to enhance the paper content. The mentioned errors were a technical issue; they were created during the file transfer as far as we were informed by the respected editor.  We sincerely apologize for the inconvenience. We hope that you receive the correct version.

Regarding the bibliography - there are very few new titles (from the last 5 years) !!

In the new bibliography, we have 32 references and 14 references are for after 2017; we hope this is acceptable by the respected reviewer.

The conclusions are not very concise and do not highlight the strengths of the article.

Yes, we agree. we enhanced the conclusion and highlighted the strengths of the article. We added two paragraphs accordingly as follows:

‘….Tire performance is the result of the interaction between the tire and the road as counter-surface. In our previous study [2, 12, 13], LAT100 was employed as a laboratory tribometer where strong correlations were obtained between the laboratory results and the tire data only on a specific counter-surface substrate. The correlations were acquired experimentally and verified and supported with statistical analyses [12, 13] and modeling [16]. Whilst the counter-surface exploited for laboratory measurements was LAT100 corundum disc 180 and seemingly different than common tire-road surfaces. The present study supports the previous work by obtaining insights into the employed substrates for laboratory and road testing….’

And the last paragraph:

…  ‘The in-depth study of the counter-surface substrates by PSD supplemented the explanations for the acquired correlations. Only disc 180 of LAT100 suggested a strong correlation for laboratory friction measurements with tire-road data in the previous studies [2, 12, 13]. The PSD graphs provided a better picture of the macro and micro texture of counter-surface substrates for the acquired correlation and highlighted the required characteristics of laboratory counter-surface substrates to evaluate tire performance. The influence of the texture wavelength range on the rubber-surface interaction could be further studied on a laboratory scale. It opens a new cost- and time-effective perspective for tire material development in a laboratory environment rather than having tested tread compounds by building full-scale tires.’

Reviewer 2 Report

The content of the paper is up to date, since vehicle-pavement interaction and tire friction are related to road safety by dramatically affecting vehicles’ stability during maneuvering. The authors used laser microscopy to assess tire texture components. The experimental analysis is well-described and the results seem rational but some further justifications are required to increase the scientific value of this paper.

  1. The introductory section of this paper (lines 25-47) should be strengthened. Research motivation is marginally given in lines 30-31. On this context, the study’s significance could be potentially improved considering that new generation cars and alternative traffic patterns (i.e. electrified cars, platooning…) could apply in the near future, necessitating the tire friction adequacy in terms of road safety. Given these issues, please refine the objective of the study (lines 45-47) to better illustrate the impact factor of the study that could affect many stakeholders (academia, industry, researchers and practitioners).
  2. Line 161: More details are needed for the asphalt sample production in order to understand its contribution to the tire friction. The characteristics of asphalt pavement surfaces (utilized materials, aggregate resistance to polishing, air voids, etc.) are detrimental for understanding the interaction between road surfaces and vehicle tires. The same also applies for concrete samples. Please elaborate.
  3. Line 159: It is unclear if replicate discs were produced to validate the retrieved results. Please comment.
  4. Dry/wet friction testers (lines 16 and 335): Please explain how wet conditions were simulated in the laboratory (i.e. water film addition, etc.) in order to understand how they could mimic real conditions in a road-vehicle environment.
  5. Conclusions: Please include potential future research plans.

Other comments include:

  1. References No. 2, 12, 13, 14, 17, 18, 31 are co-authored from the authors of the current article (7 out of 33 in total). Are all those necessary? Please consider limiting self-citations to the minimum (and absolutely necessary) extent.

Author Response

Reply to Reviewers ‘comments, Manuscript ID: lubricants-1503333

Reply to review 2

The content of the paper is up to date, since vehicle-pavement interaction and tire friction are related to road safety by dramatically affecting vehicles’ stability during maneuvering. The authors used laser microscopy to assess tire texture components. The experimental analysis is well-described and the results seem rational but some further justifications are required to increase the scientific value of this paper.

We express our gratitude for the detailed comments. We implemented the comments to enhance the context as follows:

  1. The introductory section of this paper (lines 25-47) should be strengthened. Research motivation is marginally given in lines 30-31. In this context, the study’s significance could be potentially improved considering that new generation cars and alternative traffic patterns (i.e. electrified cars, platooning…) could apply in the near future, necessitating the tire friction adequacy in terms of road safety. Given these issues, please refine the objective of the study (lines 45-47) to better illustrate the impact factor of the study that could affect many stakeholders (academia, industry, researchers, and practitioners).

The introduction was strengthened by elaborating on research motivation, the significance, and the objective of the study. We kindly refer the respected reviewer to look at the red marked line in the introduction. Moreover, a slight modification in the abstract is made to be in line with the described motivation and objective in the introduction.

  1. Line 161: More details are needed for the asphalt sample production in order to understand its contribution to the tire friction. The characteristics of asphalt pavement surfaces (utilized materials, aggregate resistance to polishing, air voids, etc.) are detrimental for understanding the interaction between road surfaces and vehicle tires. The same also applies for concrete samples. Please elaborate.

Indeed, the process of asphalt preparation includes a lot of detailed and elaborated mechanisms as the respected reviewer mentioned and it would be helpful to give the readers more information on the sample production. In the present study, the samples were taken and borrowed from Apollo Tyres to be compared with the commercialized LAT100 discs, intended for surface roughness. Unfortunately, actual production conditions are considered to be proprietary by each individual tire company, which they will not reveal in public. However, no friction measurements were performed on these asphalt and concrete surfaces, and only for the sake of comparison of tribological effect of the SURFACE with the real roads were included in the research. This minimized the need for additional information regarding the asphalt internal structure such as utilized materials, aggregate resistance to polishing, air voids, etc.

The information which we are allowed to add is that the test procedure was comparable to ASTM F1649 for measuring ABS braking on dry/wet surfaces. It is added to the experimental part of the revised manuscript and marked in red. We do hope this answers the question and satisfies the expectations of the respected reviewer.

  1. Line 159: It is unclear if replicate discs were produced to validate the retrieved results. Please comment.

We would like to kindly mention that line 159 is not referring to any disc or disc replicate, perhaps there is an error in the received version of the paper. Based on the previous comment about asphalt, we guessed that probably the respected reviewer is referring to Disc 180B and 180.

Disc 180B is not a replicate of disc 180. As described in line 170 of the old manuscript version:

...’’The letter “B” for disc 180B indicates “Blunt”, which means that the disc 180 was smoothened one step further in the finishing/flattening process using a grinding disc containing diamond powder. This smoothening process does not influence the grain size, however, it affects the surface structure and sharpness of the asperities as shown in Figure 3.’’…

In case you purchase a LAT100 device, you will receive 4 counter-surfaces discs coded 60, 180, 320, and 180B. Therefore, disc 180 and 180B are two different physical discs; only the particle size are similar but they differ in surface characteristics.

  1. Dry/wet friction testers (lines 16 and 335): Please explain how wet conditions were simulated in the laboratory (i.e. water film addition, etc.) in order to understand how they could mimic real conditions in a road-vehicle environment.

Since this paper purely focuses on surface characterizations, to not digress from the scope, we added the relevant references for this purpose. The description of dry/wet tests was added to the paper in the second line of section 2 Experimental in the revised document by referring to references 16 and 29.

Hereby, we would like to briefly explain that surface wetting in LAT100 is done using a water tube connected to the water supply that pours the water continuously on the surface. The operator can manually adjust the amount of water through a valve. Since the test configuration is mounted vertically, the water reservoir/bath as shown in the picture below guarantee that the surface remains wet in each revolution.

  1. Conclusions: Please include potential future research plans.

Yes, we agree. we enhanced the conclusion as follows in the last paragraph:

…  ‘The in-depth study of the counter-surface substrates by PSD supplemented the explanations for the acquired correlations. Only disc 180 of LAT100 suggested a strong correlation for laboratory friction measurements with tire-road data in the previous studies [2, 12, 13]. The PSD graphs provided a better picture of the macro and micro texture of counter-surface substrates for the acquired correlation and highlighted the required characteristics of laboratory counter-surface substrates to evaluate tire performance. The influence of the texture wavelength range on the rubber-surface interaction could be further studied on a laboratory scale. It opens a new cost- and time-effective perspective for tire material development in a laboratory environment rather than having tested tread compounds by building full-scale tires.’

Other comments include:

  1. References No. 2, 12, 13, 14, 17, 18, 31 are co-authored from the authors of the current article (7 out of 33 in total). Are all those necessary? Please consider limiting self-citations to the minimum (and absolutely necessary) extent.

We respect the reviewer's opinion on this. We also would like to explain in particular why these references were cited. This study of surface characterization is a supplementary work to enhance and provide answers to the remaining questions of the previous work. It is based on previous analyzing three sets of tire data in correlation with laboratory data which have already been published in the mentioned references. The references were necessary and supportive for the context and we do not consider this as self-promotion.

However, as the respected reviewer suggested, to avoid inadvertent self-citation; we removed references 14, 17, 18. To compromise, a newer paper 2021 was added which -more or less- contains the content of the removed papers in form of indications. In this way, we managed to remove two of the self-citations and fulfill the question of #reviewr 1 for newer references. We hope this helps.

Round 2

Reviewer 1 Report

I am satisfied with the changes that the authors have made as a result of the instructions made. 

Reviewer 2 Report

The manuscript was improved and there are no new comments for the authors. I guess any minor typos will be captured during proofreading.